# Atlantic deep water provenance decoupled from atmospheric $CO_2$ concentration during the lukewarm interglacials

Jacob N.W. Howe [1] & Alexander M. Piotrowski[1]

Ice core records show that atmospheric $CO_2$ concentrations and Antarctic temperature were lower during the 'lukewarm interglacials' from 800 to 430 ka than the subsequent five interglacials. These different interglacial 'strengths' have been hypothesised to be controlled by Antarctic overturning circulation. How these variations in Antarctic overturning relate to Northern Atlantic overturning circulation, a major driver of Northern Hemisphere climate, is uncertain. Here we present a high-resolution record of authigenic neodymium isotopes—a water mass tracer that is independent of biological processes—and use it to reconstruct Atlantic overturning circulation during the last 800 kyr. This record reveals a similar proportion of North Atlantic Deep Water during the 'lukewarm interglacials' and the more recent interglacials. This observation suggests that the provenance of deep water in the Atlantic Ocean can be decoupled from ventilation state of the Southern Ocean and consequently the atmospheric concentration of carbon dioxide.

---

[1] Department of Earth Sciences, University of Cambridge, Downing Street, Cambridge CB2 3EQ, UK. Correspondence and requests for materials should be addressed to J.N.W.H. (email: jacob.howe@cantab.net)

Antarctic ice core records, which span the last ~800 kyr, indicate that the glacial–interglacial cycles of the mid to late Pleistocene were not all the same in character. During the 'lukewarm interglacials' from 800 to ~430 ka, both Antarctic temperature anomaly and atmospheric carbon dioxide concentrations were significantly lower than in the interglacials of the last 430 kyr[1, 2]. The cause of the transition from 'lukewarm interglacials' to warmer interglacials around 430 ka, termed the mid-Bruhnes event (MBE)[3], is still unknown, although it has been hypothesised to be due to changes in the concentration of atmospheric carbon dioxide caused by changes in Southern Ocean structure altering the deep-to-surface exchange of $CO_2$[4]. Although sometimes thought of as a global climate event[5], the prevalence of the MBE in climate reconstructions is spatially variable and is particularly prominent in high latitude Southern Hemisphere records[1, 2, 4]. In contrast, sea surface temperature and terrestrial based temperature reconstructions from the mid-latitude North Atlantic do not show a step change at the MBE and instead show similar temperatures throughout all the interglacials of the past 800 kyr[6, 7].

Atlantic meridional overturning circulation (AMOC), which includes the formation of North Atlantic Deep Water (NADW) in the seas surrounding the North Atlantic, has an important influence on the Northern Hemisphere climate because it contributes to the meridional transport of heat[8]. Existing reconstructions of AMOC across the MBE are based on benthic foraminiferal carbon isotopes (eg, ref. [9]) that integrate changes in the carbon cycle and changes in water mass mixing[10, 11]. Such nutrient-based proxy reconstructions have therefore been interpreted in conflicting manners, with some studies inferring weaker NADW production[9, 12], while others concluded there was strong NADW production[13], during the lukewarm interglacial periods.

The neodymium isotopic ratio of seawater, expressed as $\varepsilon_{Nd}$ (deviation of $^{143}Nd/^{144}Nd$ from the chondritic uniform reservoir in parts per 10,000), is a quasi-conservative tracer of water mass source that is independent of biological processes[14]. In the modern Atlantic, $\varepsilon_{Nd}$ allows for distinction between NADW (−12.4 to −13.2) and Antarctic Bottom Water (AABW) (−8.5) and the proportion of water mass mixing between them (Fig. 1)[14–16]. Here we use neodymium isotopes measured on the authigenic phases (foraminiferal coatings and fish debris) of over 200 samples from sediment core ODP 929 from the Ceara Rise (6.0°N, 43.7°W, 4356 m; Fig. 1) to reconstruct seawater $\varepsilon_{Nd}$ in the deep equatorial western Atlantic Ocean for the last 800 kyr, yielding ~4 kyr resolution. ODP 929 sits in the mixing zone between NADW and AABW in the modern ocean (Fig. 1), thus, is sensitive to changes in water mass mixing proportions through time. Similar $\varepsilon_{Nd}$ values at Site ODP 929 occurred during all the interglacials of the past 800 kyr suggesting a similar proportion of NADW relative to AABW at ODP 929 during all of the interglacials before and after the MBE. By comparing these results with other paleoclimate reconstructions, we conclude that the provenance of water in the deep Atlantic was decoupled from the ventilation state of the Southern Ocean and consequently the atmospheric concentration of carbon dioxide.

## Results

**Authigenic $\varepsilon_{Nd}$ record of ODP 929.** The core top authigenic $\varepsilon_{Nd}$ value of ODP site 929 agrees with that of nearby seawater measurements (Fig. 2) providing confidence that this method is recording seawater $\varepsilon_{Nd}$. The $\varepsilon_{Nd}$ record of ODP 929 over the past 150 kyr has a similar structure to the smoothed record of ODP 1063 from the deep north-west Atlantic Ocean[17] over the last glacial–interglacial cycle (Fig. 3); both show the most radiogenic (positive) values during marine isotope stages (MIS) 2 and 6 and most unradiogenic (negative) values during MIS 1 and 5. The ODP 929 record does not capture the millennial scale variability seen at ODP 1063 during MIS 3 due to its lower resolution. Although the records have a similar structure, the ODP 929 $\varepsilon_{Nd}$ record is always more radiogenic than that of ODP 1063 and less radiogenic than that of RC11-83/TNO57-21 from the Cape Basin in the South Atlantic and SK129-CR2 from the Indian Ocean.

The $\varepsilon_{Nd}$ record of ODP 929 (Fig. 4a) from 800 to 0 ka shows 100-kyr glacial–interglacial cyclicity with values generally between −9.5 and −10 during glacials and more negative values around −12 to −12.5 during interglacials. These glacial–interglacial $\varepsilon_{Nd}$ cyclicity at ODP site 929 over the last 800,000 years (Fig. 4), suggest that overturning in the deep Atlantic underwent repeated transitions between similar glacial and interglacial states typified by the last glacial maximum (LGM) and the Holocene[10]. Assuming end members of approximately

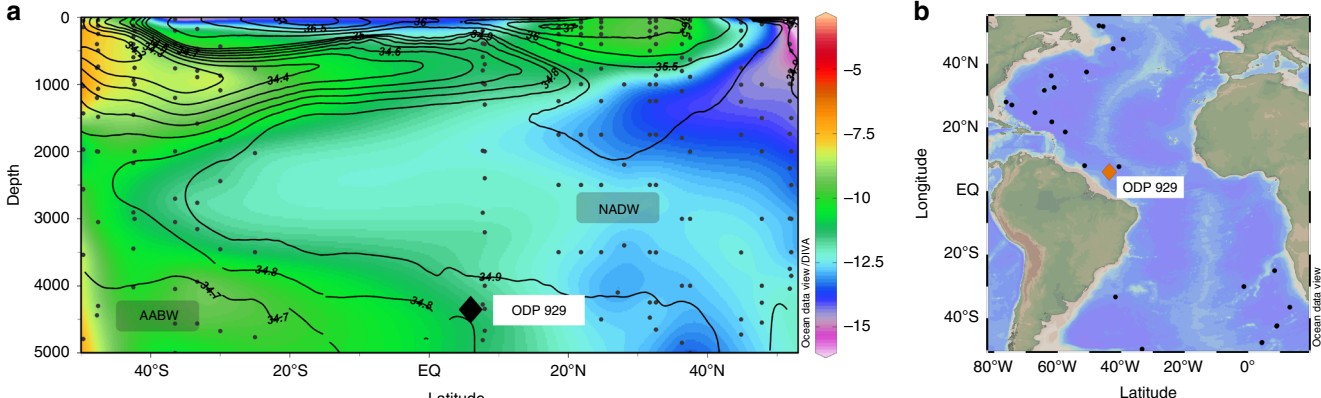

**Fig. 1** Atlantic seawater $\varepsilon_{Nd}$ profile and map showing study site location. **a** Modern seawater $\varepsilon_{Nd}$ profile for the western Atlantic Ocean and eastern Atlantic south of the Walvis Ridge[15, 16, 22, 38, 48, 49] from 55°N to 50°S with salinity contours for the western Atlantic drawn in black[50]. Data east of the mid-Atlantic ridge were excluded to remove the effect of restricted deep water communication between the eastern and western basins north of the Walvis Ridge. Data north of 55°N were excluded to remove the influence of localised deep water sources. North Atlantic Deep Water (NADW) and Antarctic Bottom Water (AABW) can be distinguished by their characteristic $\varepsilon_{Nd}$ values of −12.4 to −13.2 and −8.5, respectively. The location of ODP 929 near the boundary between NADW and AABW is plotted and labelled accordingly. **b** Map of the Atlantic Ocean showing the site (ODP 929) used in this work to measure authigenic neodymium isotopes for the past 800,000 years. Also shown in black are the location of seawater stations from which data are presented in the left panel. Figures created using Ocean Data View software (Schlitzer, R., Ocean Data View, http://odv.awi.de, 2016)

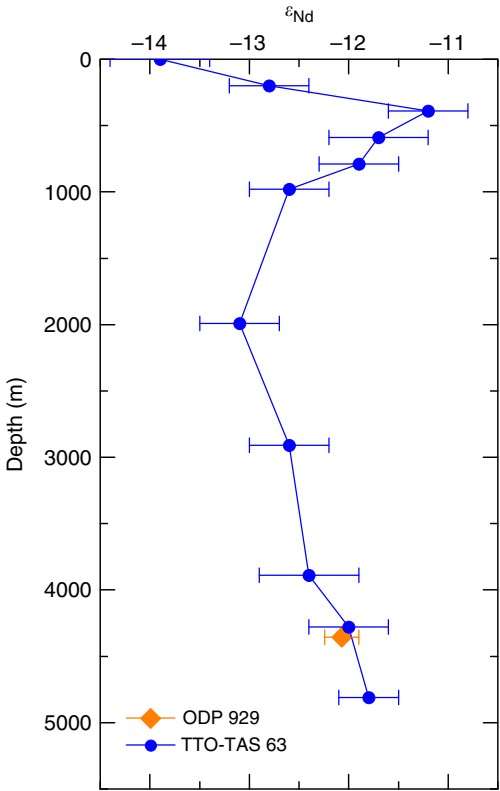

**Fig. 2** Core top foraminiferal $\varepsilon_{Nd}$ of ODP 929 compared with nearby seawater $\varepsilon_{Nd}$. ODP 929 core top planktic foraminiferal $\varepsilon_{Nd}$ vs nearby seawater $\varepsilon_{Nd}$ from station TTO-TAS 63[22]. Seawater error bars are 2$\sigma$ of the mean; foraminiferal error bars are 2$\sigma$ of the external error as calculated from the bracketing standards, unless the internal error was larger, in which case the combined external and internal error is used

−12.8 and −5.5 for northern- and southern-sourced waters, respectively, and no influence on seawater $\varepsilon_{Nd}$ from other processes (ie, benthic fluxes, particulate dissolution and so on), the data from ODP 929 are consistent with lower NADW flux during glacial periods[10]. The $\varepsilon_{Nd}$ record of ODP site 929 correlates better with benthic foraminiferal oxygen isotopes than with benthic foraminiferal carbon isotopes from the same core (Figs. 4 and 5).

## Discussion

The most straightforward interpretation of this gradient in neodymium isotopes between ODP 1063 and ODP 929 is a greater proportion of northern-sourced water at the more northerly site. ODP 929 sits at the boundary between northern- and southern-sourced water in the modern ocean (Fig. 1); therefore, would be expected to be bathed by a greater proportion of more radiogenic southern-sourced water than ODP 1063. It is worth also noting, however, that the extremely unradiogenic values from the Bermuda Rise seen during the early Holocene have been interpreted as the localised influence of unradiogenic abyssal water exported from the Labrador Sea[18]. Indeed the end-member composition of northern-sourced water during the Holocene has been shown to be more stable at mid-depths in the North Atlantic than in the deep North-west Atlantic indicating that the ODP 1063 $\varepsilon_{Nd}$ record does not represent the end-member composition of all northern-sourced water in the Atlantic. Therefore, even though ODP 929 is only ~200 m shallower than ODP 1063, the less radiogenic values observed at ODP 929 may also represent the dilution of the abyssal unradiogenic northern-sourced water seen at ODP 1063

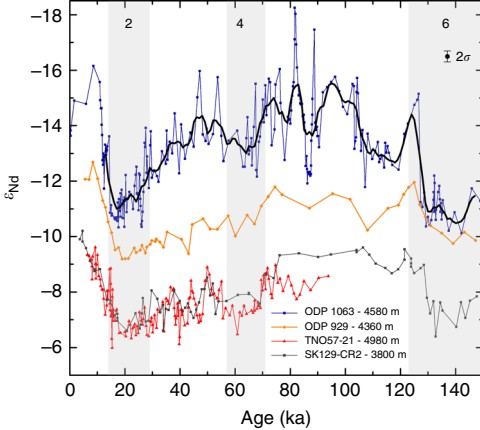

**Fig. 3** Comparison of authigenic $\varepsilon_{Nd}$ records from the Atlantic and Indian Oceans. Authigenic $\varepsilon_{Nd}$ spanning the last 150 kyr for ODP 1063/OCE326-6GGC (blue)[17, 44], with a five-point smoothed running mean of the 1 kyr linearly interpolated record (black), with $\varepsilon_{Nd}$ of ODP 929 (orange), TNO57-21/RC11-83 (red)[19, 51], and SK129-CR2 (grey)[20, 21]. 2$\sigma$ shows the average error for all data shown, as reported in the corresponding publications and this work. Glacial marine isotope stages are shaded in grey and labelled with their corresponding numbers

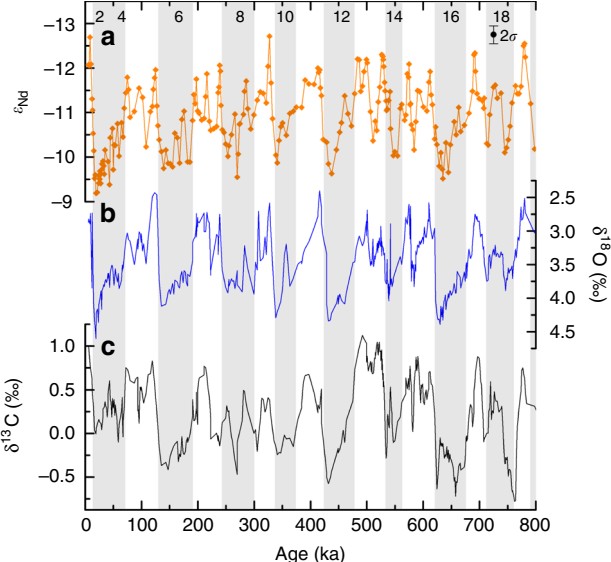

**Fig. 4** Authigenic $\varepsilon_{Nd}$, benthic foraminiferal $\delta^{18}O$ and $\delta^{13}C$ of ODP 929. Down core records from western equatorial Atlantic ODP site 929: (**a**) authigenic neodymium isotopes ($\varepsilon_{Nd}$) measured on planktic foraminifera and fish debris. 2$\sigma$ shows the average external error for all data shown except in cases where the internal error was larger in which case the combined external and internal error was used. **b** Benthic foraminiferal oxygen isotopes[42]. **c** Benthic foraminiferal carbon isotopes smoothed with a three-point running mean[42]. Glacial marine isotope stages are shaded in grey and numbered accordingly

with more radiogenic northern-sourced water from the mid-depth Atlantic[18]. The observation that the unradiogenic values observed at ODP 1063 are restricted to the abyssal Atlantic is, however, limited to Holocene data and although we assume it to be true of earlier warm periods for the purposes of this discussion, more data would increase the certainty of this assumption.

Notwithstanding these end-member uncertainties, the more radiogenic values of TNO57-21/RC11-83 than ODP 929 (Fig. 3)

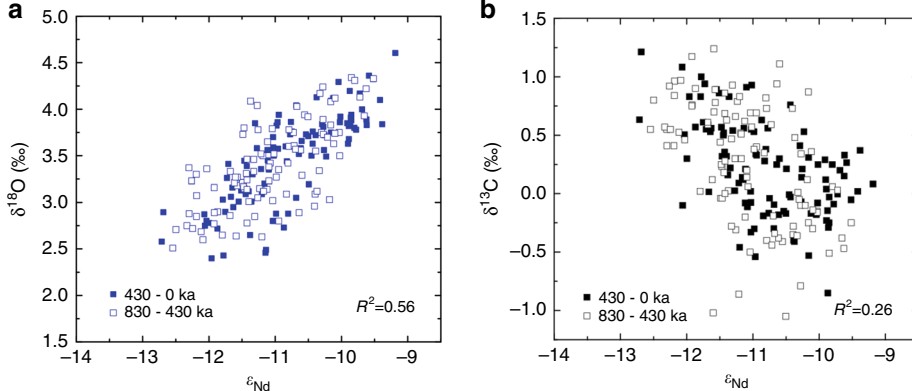

**Fig. 5** Authigenic $\varepsilon_{Nd}$ vs benthic foraminiferal $\delta^{18}O$ and $\delta^{13}C$ of ODP 929. Cross plots of authigenic $\varepsilon_{Nd}$ against (**a**) benthic foraminiferal $\delta^{18}O$ and (**b**) benthic foraminiferal $\delta^{13}C$ all from ODP 929 from 0.8 to 0 Ma with the $R^2$ value for the correlation between the cross-plotted variables given in each plot

always indicate a lesser proportion of northern-sourced water at the more southerly site of TNO57-21/RC11-83, which sits at the boundary between the South Atlantic and Southern Ocean[19]. A lesser proportion of northern-sourced water than ODP 929 can also be inferred from the $\varepsilon_{Nd}$ record from SK129-CR2, from 3800 m water depth in the equatorial Indian Ocean (Fig. 3) that records the composition of deep water exported from the Southern Ocean into the Indian Ocean[20, 21]. The more radiogenic values of the South Atlantic Cape Basin and Indian Ocean records relative to ODP 929 supports the inference of sustained production of NADW during the interglacials and glacials of the last 150 kyr, irrespective of the influence of Labrador Sea Water at ODP 1063.

ODP 929 displays $\varepsilon_{Nd}$ values within error of the modern sea-water value at that site (−12.1; ref. [22]) during all of the interglacials of the past 800 kyr (Fig. 4a), implying that northern-sourced water was equally dominant in the deep Atlantic during all of the peak interglacial periods of the last 800 kyr. This conclusion contrasts with the findings of some studies based on benthic foraminiferal carbon isotopes[9, 12] but agrees with others[13]. Our conclusions disagree with the findings of studies that inferred the expanse of northern-sourced water in the deep ocean was less during lukewarm interglacials by comparing benthic foraminiferal $\delta^{13}C$ records from the Atlantic and Pacific[9, 12]. Instead, we suggest that this difference is likely to due to the alteration of the $\delta^{13}C$ of AABW by processes such as air–sea exchange during deep water formation and nutrient regeneration along deep water flow[21]. The strong correlation of $\varepsilon_{Nd}$ with benthic foraminiferal $\delta^{18}O$—which represents a combined signal of deep water temperature and global ice volume—suggests a tight coupling between global ice volume, NADW production and Atlantic deep water temperature at orbital time scales. The weaker correlation between $\varepsilon_{Nd}$ and benthic foraminiferal $\delta^{13}C$, both water mass tracers, is most likely attributable to long-term global shifts in the carbon cycle[11], air–sea exchange in the Southern Ocean[21] and short-term carbon cycle processes within the Atlantic, such as the respiration of organic matter[10], decoupling $\delta^{13}C$ from water mass mixing.

Just as possible changes in northern-sourced deep water $\varepsilon_{Nd}$ composition have been documented in the deep north-west Atlantic for the Holocene[17, 18], it must also be considered that the neodymium composition of northern-sourced water and the proportion of different northern-sourced waters (such as Iceland Straights Overflow Waters and Labrador Sea Water) may have varied during the glacial–interglacial cycles of the past 800 kyr. High-resolution Fe-Mn oxide crusts from the intermediate to mid-depth Atlantic show little variation in composition across the

past 500 kyr including MIS 13, a lukewarm interglacial that precedes the MBE[23]. This suggests the northern-sourced end member was relatively constant over that time period, arguing against any major increase in the proportion of less radiogenic Labrador Sea Water at the expense of other, more radiogenic, northern-sourced waters.

Assuming that deep northern-sourced water did have similar neodymium compositions across all of the interglacials of the past 800 kyr suggests that northern-sourced water dominated the deep Atlantic during the lukewarm interglacials as much as it did in the more recent interglacials. Combining this inference of water mass provenance with evidence of strong deep water circulation and flowspeeds[13], implies that NADW was as vigorous, and as extensive during the lukewarm interglacials as it is in the present interglacial climate. This inference of strong NADW production along with the presence of small Northern Hemisphere ice sheets —inferred from sea level reconstructions (Fig. 6b)—during the lukewarm interglacials despite the 30–40 p.p.m. lower atmospheric $CO_2$ concentrations than during the more recent interglacials (Fig. 6) reveals that the lower atmospheric carbon dioxide concentrations during those intervals were unlikely to be related to Northern Hemisphere processes. Indeed, the lower atmospheric $CO_2$ concentrations during the lukewarm interglacials than more recent interglacial has been attributed to smaller dynamic range of Antarctic overturning[4], suggesting a closer link to Southern Hemisphere, and Southern Ocean, processes. This smaller dynamic range of Antarctic overturning during lukewarm interglacials was inferred from reconstructions of $CaCO_3$ preservation and export flux made using the Ca/Fe (Fig. 6c) and Ba/Fe measurements from site ODP 1094 in the Antarctic zone of the Southern Ocean. The lack of $CaCO_3$ preservation peaks during the lukewarm interglacials (Fig. 6c) has been interpreted to be the result of poorer deep-to-surface exchange of water in the Southern Ocean during the lukewarm interglacials[4]. This was deemed to be one of the two modes of Southern Ocean productivity responsible for the structure of the atmospheric $CO_2$ record (Fig. 6d), the second being iron fertilisation of the sub-Antarctic zone of the Southern Ocean[4].

The conclusion drawn here that NADW production was as strong during the lukewarm interglacials as during recent interglacials, including the present interglacial, implies that during each deglaciation, from 830 ka to the present, NADW production increased from a glacial state to the same full interglacial state. In contrast, Southern Ocean ventilation—inferred from preservation records (Fig. 6c)—was different during the lukewarm interglacials relative to more recent interglacials[4]. It is therefore evident that the lukewarm interglacials correspond to a climate state with a

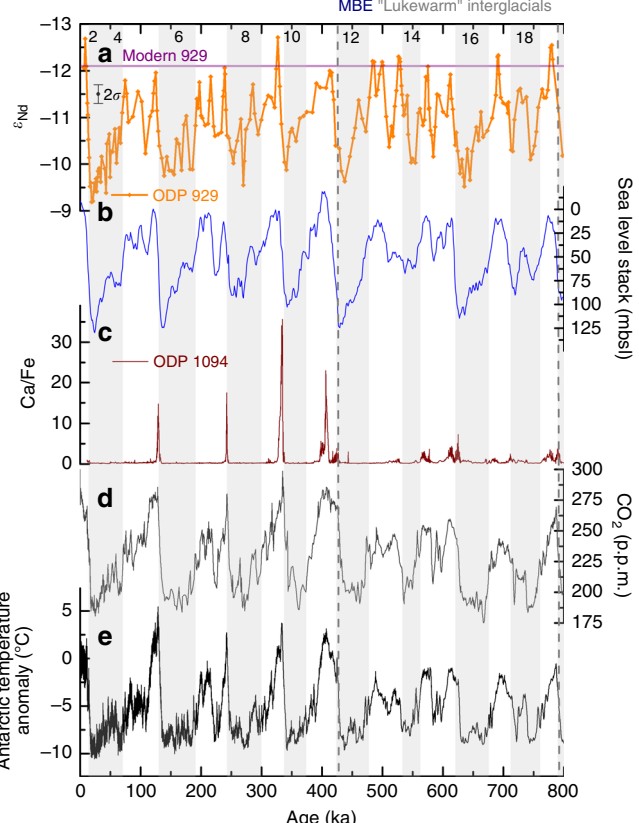

**Fig. 6** Authigenic $\varepsilon_{Nd}$ of ODP 929 compared with climate reconstructions for the past 800 kyr. **a** Authigenic $\varepsilon_{Nd}$ of ODP site 929 with modern seawater $\varepsilon_{Nd}$ at that site denoted by the purple line. $2\sigma$ shows the average external error for all data shown except in cases where the internal error was larger in which case the combined external and internal error was used. **b** Stack of sea level reconstructions[52]. **c** Five-point smoothed running mean of Ca/Fe from XRF of ODP 1094, a proxy for dynamic range of Antarctic overturning (53.2°S, 05.1°W, 2850 m)[4]. Composite Antarctic ice core records of: (**d**) atmospheric $CO_2$ concentration[2] and (**e**) Antarctic temperature anomaly[1]. The mid-Brunhes event (MBE) and the preceding 'lukewarm' interglacials are labelled and the latter are highlighted by the dashed grey box; glacial marine isotope stages are shaded in grey and numbered

high proportion of northern-sourced water and likely strong NADW production, but a poorly ventilated deep Southern Ocean. Therefore, we conclude that the North Atlantic- and Antarctic-derived deep ocean overturning circulation cells were decoupled from one another because full resumption of NADW production during deglaciations prior to the MBE did not result in ventilation of the deep Southern Ocean comparable to that seen during later interglacials (Fig. 6c).

It is clear from our results that the poorer Southern Ocean ventilation, lower atmospheric $CO_2$ and lower Antarctic temperatures of the lukewarm interglacials did not impede NADW production (Fig. 6), which was as strong as during recent interglacials. The exchange of warm surface waters from the Indian Ocean to the South Atlantic via Agulhas leakage is thought to be an essential component of NADW formation[24]. The fronts in the Atlantic sector of the Southern Ocean were further northward during the lukewarm interglacials than more recent interglacials[25]; however, whether this frontal shift impeded Agulhas leakage is unclear. While some reconstructions infer weaker Agulhas leakage during the lukewarm interglacials[12], others suggest there was strong Agulhas leakage during the most recent

Lukewarm interglacial, MIS 13[24]. The location of the fronts further north during the lukewarm interglacials may have been the key to how the ventilation state of the Southern Ocean, and as a consequence atmospheric $CO_2$ concentration, were decoupled from North Atlantic climate changes during the lukewarm interglacials. It has been proposed that shifts in the winds around Antarctica played an important role in isolating warm northern-sourced waters from Antarctica during deglaciation until an abrupt change in atmospheric circulation triggered complete deglaciation[26]. This wind-shift mechanism combined with the evidence that the fronts in the Southern Ocean were further north during the lukewarm interglacials reconciles our inference of strong NADW production during the lukewarm interglacials despite lower Antarctic temperatures and atmospheric $CO_2$ concentrations. The heat released by upwelling NADW in the Southern never reached Antarctica as it was thermally isolated by the northerly location of the fronts in the Atlantic sector of the Southern Ocean.

We propose that strong NADW production during the lukewarm interglacials may therefore have been sustained by Agulhas leakage and perhaps by strong Antarctic Intermediate Water (AAIW) formation. Radiocarbon evidence indicates that the intermediate depth South Atlantic was well ventilated, during the LGM[27, 28] with stable isotope evidence[29] revealing that this ventilation was achieved by AAIW. The strong production of AAIW under glacial conditions suggests that cooler Antarctic temperatures during the lukewarm interglacials did not reduce AAIW formation. Meanwhile, warmer climate conditions in the Northern Hemisphere during the lukewarm interglacials than during the preceding or following glacial periods[30] may have preferentially favoured strong NADW formation rather than the strong Glacial North Atlantic Intermediate Water formation commonly inferred to have occurred during the most recent glacial period[31, 32].

In the Southern Ocean, AAIW is formed north of the Polar Front, whereas AABW is formed through shelf processes and deep convection south of the Polar Front[33], therefore these two southern-sourced water masses need not be dynamically controlled by the same climatic variations. It is however, important to note that while the ventilation of the deep Southern Ocean was different during the lukewarm interglacials compared to more recent interglacials[4], this does not necessarily imply that AABW circulation was slower in the deep ocean, indeed it is often assumed that AABW formation was stronger during glacial periods[34]. It does suggest the ventilation of the deep water to the atmosphere was less effective, either because of greater sea ice caused by cooler temperatures around Antarctica or increased stratification of the Southern Ocean[35, 36] linked to stronger AABW production.

Our results reveal that $\varepsilon_{Nd}$ values in the equatorial Atlantic were similar during the lukewarm and more recent interglacials (Fig. 4), while other $\varepsilon_{Nd}$ reconstructions from the North Atlantic[23] and the South Atlantic[37] suggest that the end-member water mass compositions were similar to modern values during the lukewarm interglacials[15, 38] implying a similar water mass provenance at site ODP 929 in the deep equatorial Atlantic throughout all of the interglacials of the past 800 kyr. Given the flow speed proxy evidence from the North Atlantic[13] for strong bottom water flows in the North Atlantic during the lukewarm interglacials, in order to compensate and achieve similar water mass proportions at site ODP 929 requires that during the lukewarm interglacials southern-sourced water must have flowed northwards in the Atlantic at similar strength as more recent interglacials. This suggests that there was no difference in Antarctic overturning due to the rate of AABW production between the lukewarm and more recent interglacials, so that the primarily

climate effect was not a difference in physical overturning but the efficiency of exchange of dissolved carbon with the atmosphere in the Southern Ocean. Flow speed proxy data from the South Atlantic during the lukewarm interglacials would help to further address this question.

As discussed earlier, our results can be reconciled with existing climate reconstructions by invoking a disconnection between Antarctic and sub-Antarctic climate caused by position of the westerly winds and accordingly the fronts in the Southern Ocean[26, 39]. As the circum-Antarctic region was colder during the lukewarm interglacials than more recent interglacials, it follows that there may have been a greater extent of sea ice during the lukewarm interglacials. This greater sea ice extent, along with possible greater stratification of the Southern Ocean, could explain how AABW production was able to be strong during the lukewarm interglacials but also result in a poorly ventilated water mass. The ventilation of the Southern Ocean being responsible for the 30–40 p.p.m. difference between the lukewarm and more recent interglacials agrees well with a box model study of carbons storage mechanisms in the deep ocean[36].

It is important to note that the conclusion made here that NADW production was decoupled from atmospheric carbon dioxide concentration during the lukewarm interglacials applies only to that specific 30–40 p.p.m. difference between the lukewarm interglacials and more recent interglacials that has been attributed to changes in Southern Ocean ventilation[4, 36]. This finding does not preclude the influence of both NADW production and water mass provenance in the deep Atlantic upon deep ocean carbon storage that has been proposed for other time periods such as the LGM[10].

In light of recent changes observed in North Atlantic overturning under a warming interglacial climate[40] and given the influence of Southern Ocean on atmospheric $CO_2$ concentration[4], it is vital that we understand how Atlantic overturning changes effect Antarctic overturning. Our results reveal that changes in North Atlantic-derived overturning have not always been coincident with changes in the ventilation state of the Southern Ocean nor atmospheric $CO_2$ concentration, and instead were operating independently from the global ocean circulation.

## Methods

**Site and age model**. ODP site 929 (6.0°N, 43.7°W, 4356 m), was retrieved during ODP Leg 154 from the Ceara Rise in the equatorial western Atlantic. A single continuous splice record, constructed from holes A-D avoiding sections of significant disturbance or distortion, was published in the coring report[41]; the average sedimentation rate throughout the Pleistocene section is 4 cm per kyr. The published benthic foraminiferal $\delta^{18}O$ and $\delta^{13}C$ records from this core have two prominent gaps, across MIS 10 and 11 and in MIS 13[42]. These gaps were filled in this work using the same species as used in the published record. The Pleistocene age model of ODP 929 was updated by tying the newly completed benthic foraminiferal $\delta^{18}O$ record to the LR04 stack[43]. The deglacial and Holocene section was taken from the B-hole due to a lack of availability of samples from the A-hole used in the original splice section. The B-hole samples were dated using radiocarbon measurements of *Globigerinoides sacculifer* combined with *Globigerinoides ruber* where required (Supplementary Data 1 and ref. [18]), and then tied to the splice record using both the benthic foraminiferal $\delta^{18}O$ and planktic radiocarbon measurements.

**Stable isotopes**. *Cibicidoides wuellerstorfi*, mixed *Cibicidoides* species or *Nuttallides umbonifera*, were picked from the coarse (>63 µm) fraction of samples from MIS 10, 11 and 13 dependent upon species availability. Samples were analysed by the Godwin Laboratory at the University of Cambridge using a Micromass Multicarb Sample Preparation System attached to either a VG SIRA or a Thermo Kiel device attached to a Thermo MAT253 Mass Spectrometer in dual inlet mode. Isotopic ratios are presented relative to standard Vienna PeeDee Belemnite; external precision was ±0.06‰ for $\delta^{13}C$ and ±0.08‰ for $\delta^{18}O$. The new foraminiferal stable isotope data is provided in Supplementary Data 2. A species corrections factor of −0.2‰ to $\delta^{18}O$ and +0.2‰ to $\delta^{13}C$ was applied to results obtained from *N. umbonifera* before plotting[42].

**Neodymium separation**. From 0.8–0 Ma, the composite splice was sampled for authigenic neodymium on average every 15 cm, giving ~4 kyr resolution. Foraminifera samples were prepared for analysis following the methods of ref. [44]. In brief, where possible, up to 80 mg of mixed planktic foraminifera were picked from the coarse fraction (>63 µm) for neodymium isotope measurements. Following crushing and sonication for clay removal, samples were dissolved in 1 mol L$^{-1}$ reagent grade acetic acid. Where sufficient foraminifera were not available, fish debris were picked from the coarse fraction (>63 µm); fish debris samples were dissolved in 3 mol L$^{-1}$ qHNO$_3$ with sonication. Chemical cleaning of the fish debris was deemed unnecessary as cleaned and uncleaned samples have been shown to yield the same results[45]. The detrital fractions of four samples were also prepared for neodymium isotopic analysis following the procedures in ref. [46]. Following dissolution of the given phases, rare earth elements were extracted using Eichrom TRUspec™ resin. Neodymium was then isolated from the other rare earth elements using Eichrom LNspec™ resin.

**Neodymium isotopic analysis**. Neodymium isotopes were analysed using the Nu Plasma HR or Neptune Plus multi-collector inductively coupled plasma mass spectrometers at the University of Cambridge.$^{146}Nd/^{144}Nd$ was normalised to 0.7219 using an exponential mass fractionation correction and samples were bracketed with a concentration-matched solution of reference standard JNdi-1, the measured composition of which varied between runs but was corrected to the accepted value of $^{143}Nd/^{144}Nd = 0.512115$[47]. The $\varepsilon_{Nd}$ of each sample is reported in Supplementary Data 3 with the standard deviation ($2\sigma$) of the bracketing standards from the corresponding measurement session (external error), unless the internal error was larger than the external error, in which case the combined (square root of the sum of squares) internal and external error ($2\sigma$) is reported.

**Data availability**. The data reported in this paper is provided in supplementary information and archived in Pangaea (https://doi.pangaea.de/10.1594/PANGAEA.882970).

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

# ARTICLE

16. Stichel, T., Frank, M., Rickli, J. & Haley, B. A. The hafnium and neodymium isotope composition of seawater in the Atlantic sector of the Southern Ocean. *Earth Planet. Sci. Lett.* **317–318**, 282–294 (2012).

17. Böhm, E. et al. Strong and deep Atlantic meridional overturning circulation during the last glacial cycle. *Nature* **517**, 73–76 (2015).

18. Howe, J. N. W., Piotrowski, A. M. & Rennie, V. C. F. Abyssal origin for the early Holocene pulse of unradiogenic neodymium isotopes in Atlantic seawater. *Geology* **831**, 831–834 (2016).

19. Piotrowski, A. M., Goldstein, S. L., Hemming, S. R. & Fairbanks, R. G. Temporal relationships of carbon cycling and ocean circulation at glacial boundaries. *Science* **307**, 1933–1938 (2005).

20. Piotrowski, A. M. et al. Indian Ocean circulation and productivity during the last glacial cycle. *Earth Planet. Sci. Lett.* **285**, 179–189 (2009).

21. Wilson, D. J., Piotrowski, A. M., Galy, A. & Banakar, V. K. Interhemispheric controls on deep ocean circulation and carbon chemistry during the last two glacial cycles. *Paleoceanography* **30**, PA2707 (2015).

22. Piepgras, D. J. & Wasserburg, G. J. Rare earth element transport in the western North Atlantic inferred from Nd isotopic observations. *Geochim. Cosmochim. Acta* **51**, 1257–1271 (1987).

23. Foster, G. L., Vance, D. & Prytulak, J. No change in the neodymium isotope composition of deep water exported from the North Atlantic on glacial-interglacial time scales. *Geology* **35**, 37–40 (2007).

24. Beal, L. M., De Ruijter, W. P. M., Biastoch, A. & Zahn, R. On the role of the Agulhas system in ocean circulation and climate. *Nature* **472**, 429–436 (2011).

25. Kemp, A. E. S., Grigorov, I., Pearce, R. B. & Naveira Garabato, A. C. Migration of the Antarctic polar front through the mid-Pleistocene transition: evidence and climatic implications. *Quat. Sci. Rev.* **29**, 1993–2009 (2010).

26. Adkins, J. F. The role of deep ocean circulation in setting glacial climates. *Paleoceanography* **28**, 539–561 (2013).

27. Sortor, R. N. & Lund, D. C. No evidence for a deglacial intermediate water $\Delta^{14}$C anomaly in the SW Atlantic. *Earth Planet. Sci. Lett.* **310**, 65–72 (2011).

28. Freeman, E. et al. An Atlantic–Pacific ventilation seesaw across the last deglaciation. *Earth Planet. Sci. Lett.* **424**, 237–244 (2015).

29. Tessin, A. C. & Lund, D. C. Isotopically depleted carbon in the mid-depth South Atlantic during the last deglaciation. *Paleoceanography* **28**, 296–306 (2013).

30. Candy, I. et al. Pronounced warmth during early middle Pleistocene interglacials: investigating the mid-Brunhes event in the British terrestrial sequence. *Earth Sci. Rev.* **103**, 183–196 (2010).

31. Curry, W. B. & Oppo, D. W. Glacial water mass geometry and the distribution of $\delta^{13}$C of $\Sigma CO_2$ in the western Atlantic Ocean. *Paleoceanography* **20**, PA1017 (2005).

32. Ferrari, R. et al. Antarctic sea ice control on ocean circulation in present and glacial climates. *Proc. Natl. Acad. Sci. USA* **111**, 8753–8758 (2014).

33. Meredith, M. P. et al. Distribution of oxygen isotopes in the water masses of Drake Passage and the South Atlantic. *J. Geophys. Res.* **104**, 20949–20962 (1999).

34. Govin, A. et al. Evidence for northward expansion of Antarctic Bottom Water mass in the Southern Ocean during the last glacial inception. *Paleoceanography* **24**, PA1202 (2009).

35. Sigman, D. M., Hain, M. P. & Haug, G. H. The polar ocean and glacial cycles in atmospheric $CO_2$ concentration. *Nature* **466**, 47–55 (2010).

36. Hain, M. P., Sigman, D. M. & Haug, G. H. Carbon dioxide effects of Antarctic stratification, North Atlantic intermediate water formation, and subantarctic nutrient drawdown during the last ice age: diagnosis and synthesis in a geochemical box model. *Global Biogeochem. Cycles* **24**, GB4023 (2010).

37. Pena, L. D. & Goldstein, S. L. Thermohaline circulation crisis and impacts during the mid-Pleistocene transition. *Science* **345**, 318–322 (2014).

38. Jeandel, C. Concentration and isotopic composition of Nd in the South Atlantic Ocean. *Earth Planet. Sci. Lett.* **117**, 581–591 (1993).

39. Anderson, R. F. et al. Wind-driven upwelling in the Southern Ocean and the deglacial rise in atmospheric $CO_2$. *Science* **323**, 1443–1448 (2009).

40. Bryden, H. L., Longworth, H. R. & Cunningham, S. A. Slowing of the Atlantic meridional overturning circulation at 25°N. *Nature* **438**, 655–657 (2005).

41. Curry, W. B., Shackleton, N. J. & Richter, C. Site 929. *Proc. Ocean Drill. Prog. Initial Rep.* **154**, 337–413 (1995).

42. Bickert, T., Curry, W. & Wefer, G. Late Pliocene to Holocene (2.6-0 Ma) western equatorial Atlantic deep-water circulation: inferences from benthic stable isotopes. *Proc. Ocean Drill. Prog. Sci. Results* **154**, 239–254 (1997).

43. Lisiecki, L. E. & Raymo, M. E. A Pliocene-Pleistocene stack of 57 globally distributed benthic $\delta^{18}$O records. *Paleoceanography* **20**, PA1003 (2005).

44. Roberts, N. L., Piotrowski, A. M., McManus, J. F. & Keigwin, L. D. Synchronous deglacial overturning and water mass source changes. *Science* **327**, 75–78 (2010).

45. Martin, E. E. et al. Extraction of Nd isotopes from bulk deep sea sediments for paleoceanographic studies on Cenozoic time scales. *Chem. Geol.* **269**, 414–431 (2010).

46. Noble, T. L. et al. Greater supply of Patagonian-sourced detritus and transport by the ACC to the Atlantic sector of the Southern Ocean during the last glacial period. *Earth Planet. Sci. Lett.* **317–318**, 374–385 (2012).

47. Tanaka, T. et al. JNdi-1: a neodymium isotopic reference in consistency with LaJolla neodymium. *Chem. Geol.* **168**, 279–281 (2000).

48. Garcia-Solsona, E. et al. Rare earth elements and Nd isotopes tracing water mass mixing and particle-seawater interactions in the SE Atlantic. *Geochim. Cosmochim. Acta.* **125**, 351–372 (2014).

49. Huang, K.-F., Oppo, D. W. & Curry, W. B. Decreased influence of Antarctic intermediate water in the tropical Atlantic during North Atlantic cold events. *Earth Planet. Sci. Lett.* **389**, 200–208 (2014).

50. Antonov, J. I. et al. *World Ocean Atlas 2009, Volume 2: Salinity.* (U.S. Government Printing Office, Washington, D.C., 2010).

51. Piotrowski, A. M., Goldstein, S. L., Hemming, S. R. & Fairbanks, R. G. Intensification and variability of ocean thermohaline circulation through the last deglaciation. *Earth Planet. Sci. Lett.* **225**, 205–220 (2004).

52. Spratt, R. M. & Lisiecki, L. E. A late Pleistocene sea level stack. *Clim. Past* **12**, 1079–1092 (2016).

## Acknowledgements
Sample material was provided by the International Ocean Drilling Program. D. Wilson, S. Crowhurst and A. Elmore are thanked for comments on an earlier version of this manuscript. S. Williams, J. Clegg, V. Rennie, S. Misra, J. Day, E. Tipper and H. Chapman are thanked for technical support. T. Bickert is thanked for providing published data from ODP site 929. Radiocarbon analyses were supported by NERC radiocarbon grant 1752.1013 and Nd isotope analyses by NERC grants NE/K005235/1 and NE/F006047/1 to A.M.P. J.N.W.H. was supported by a Rutherford Memorial Scholarship.

## Author contributions
J.N.W.H. and A.M.P. designed the study, J.N.W.H. performed the work, and J.N.W.H. and A.M.P. wrote the manuscript.

## Additional information

**Competing interests:** The authors declare no competing financial interests.

