## [Peer Review File · Nature Communications]

Reviewers' comments:

Reviewer #1 (Remarks to the Author):

Jake Howe and Alex Piotrowski present a beautifully resolved, very robust eNd record from the eastern tropical Atlantic (ODP site 929). The record is quite unique and certainly represents an important step forward in our understanding of changes in ocean circulation through time. The location of the core seems to robustly record temporal changes in the proportion of northern- vs. southern sourced waters and its core top value aligns well with water-column measurements. The record reveals that the proportion of northern sourced water remained quite stable during the interglacials of the past 800 kyrs. This observation is used to infer that the deep ocean overturning circulation sourced in the North Atlantic was independent upon the state of Antarctic overturning in the Southern Hemisphere during the lukewarm interval (430-800 kyrs). Jaccard et al., 2013 had previously suggested that the relatively colder lukewarm interglacials were characterized by a reduced dynamic range of overturning, which according to the authors did not affect deep water production in the North Atlantic. As such, they propose that both deep convection end-members were characterized by independent tipping points. This is certainly an interesting hypothesis, but to my opinion the argument developed throughout the manuscript is somewhat overstated given the evidence at hand. At this stage the manuscript remain quite descriptive and would benefit from a much more inclusive discussion.

Major comments –

* tipping point – I find this formulation unfortunate and quite frankly confusing. Many studies (e.g. Ramstorf, 2002; Böhm et al., 2015) have shown that AMOC intensity was evolving along a continuum. The only notable change in AMOC regime was reported to be associated with Heinrich stadials (off mode).

The core site was always bathed by both AABW and NADW in various proportions as beautifully illustrated by the downcore record. Moreover, the comparison between d18O and eNd (Fig. S2a) suggests a coherent correlation throughout the interval considered.

Jaccard et al., 2013 invoke a reduced dynamic range of Antarctic overturning during the lukewarm interglacials, which I interpret as representing a continuum as well. The much more subtle wording used in the abstract (l. 21-23) is much more convincing and appropriate.

Furthermore, Adkins, 2013 argues for a strong coupling between northern- and southern overturning circulation, in contradiction with the hypothesis presented here (e.g. l. 113-117). The argumentation would generally benefit a lot from a more inclusive discussion on these two particular points.

* I find it very surprising that the authors omit to compare their results with the Böhm et al., 2015 ODP1063 record covering the interval 0-150 kyrs. Both records share many similarities, but are also characterized by substantial differences, notably during MIS 5e and 6 (this also applies to the eNd results published recently by Deaney et al., 17). These discrepancies could be related to the (relatively) lower resolution of the record presented here, which could potentially undermine the conclusions presented in the manuscript.

* Can the authors exclude a scenario for which NADW would be strongly reduced during the lukewarm interglacial and would be replaced by stronger Labrador Sea water (LSW)? One could imagine a switch in the way AMOC is being circulated during the lukewarm interval, during which LSW would provide a much larger contribution than today.

The more general point I want to make here is that, taken at face value, the data only robustly indicate that a variable proportion of LSW was present at the core site throughout the past 800 kyrs (given the quite unradiogenic values overall) without allowing making any inference about NADW (*sensu stricto*). I would thus recommend using the more generic term “northern-sourced waters” instead of NADW.

Minor comments –

I. 29 – "... Pleistocene WERE not all..."

I.82–85 & Fig. S2a – It would perhaps be a good exercise to distinguish the data from the lukewarm interval and see whether the eNd-d18O correlation is characterized by a different slope when compared with the last 430 kyrs.

I.95-96 – "... reveals that atmospheric carbon dioxide concentrations variation DURING THIS PARTICULAR INTERVAL is unlikely to be related to Northern..."

I.109 – "... was different DURING the lukewarm..."

I.122 and throughout – "Agulhas" not "Agulhus"

I.130 – AABW mostly forms through shelf processes and deep convection. The fraction of AABW flowing into the Atlantic Basin mostly forms in the Weddell Sea.

I. 133-136. Very interesting argument!

I. 137 – eNd does not allow making any inference about flow speed/strength. Please rephrase.

I. 146-152. The conclusion is overstated and largely speculative. I would recommend removing these few sentences.

I. 179 - "are" not "ate"

I. 279 - what type of relationship has been used to correct for the mass bias (0.7219) - exponential?

I. 282 – Are the authors referring to the standard error or the standard deviation?

Reviewer #2 (Remarks to the Author):

In their paper 'Independent tipping points for North Atlantic and Antarctic overturning circulation' Howe and Piotrowski present a new high resolution record of authigenic neodymium (Nd) isotopes from the deep equatorial western Atlantic Ocean spanning the past 800,000 years. The authors interpret the reoccurrence of similar Nd isotopic compositions in glacial and interglacial periods throughout the record to indicate similar proportions of vigorous NADW flow at the site (i.e. full NADW resumption) at each of the interglacials. This observation is in turn taken to indicate decoupling of Northern Hemisphere processes from atmospheric carbon dioxide variations, which are 30-40 ppm lower during the lukewarm interglacials in the older parts of the record. In conjunction with a previously published productivity record from the Antarctic Zone in the Southern Ocean indicating reduced Antarctic overturning during lukewarm interglacials, the authors conclude that there are different tipping points for North Atlantic and Antarctic overturning circulations.

This new record is an exciting addition to develop our understanding on ocean circulation over the past 800,000 years and its role in setting atmospheric carbon dioxide concentrations. As such it should definitely be published. For my liking the paper however is too brief. As a consequence it leaves the reader a bit in the dark on the proposed mechanisms of 'independent tipping points for North Atlantic and Antarctic overturning circulation'. I will provide some examples below on points that could be further developed.

(1) Quantitative statements on NADW flux

In lines 66 to 70 the authors use their own, previously published and carefully documented estimates for glacial endmember compositions of northern and southern-sourced waters to conclude that the new data from ODP Site 929 are consistent with lower NADW flux during glacial periods. This is in agreement with previous work (e.g. Bohm et al., 2015, Nature) but has not been demonstrated for the duration of the past 800,000 years and is hence an important finding. Suggested similar proportions of NADW during peak interglacials indirectly suggest that we are looking at two sets of endmember compositions for the Southern Ocean for modern and glacial conditions, which means that it is rather difficult to say something about the rigour or flux of water masses based on Nd isotopes alone. I am also wondering how the authors reconcile the apparent stability in northern endmember composition with their own findings of changed early Holocene NADW values in the abyssal ocean (Howe et al., 2016, Geology) in the light of such anomalous low values occurring at various points during the last glacial cycle (e.g. Boehm et al., 2015, Nature). I am not necessarily disagreeing on the point they make, but it would be great to see a more detailed discussion on how some of the interpretations can be reconciled in terms of modern and past Nd isotope data and endmembers.

(2) North Atlantic and Southern Ocean overturning circulation

Along the same lines, it would be nice to get a bit more detail on what is actually meant by North Atlantic and Southern Ocean overturning circulations. Are we talking about the upper and lower limb of the overturning circulation in the way as defined by Ferrari et al. (2014, PNAS)? For the North Atlantic the answer is most certainly yes, but what about the Southern Ocean? The Antarctic overturning circulation in Jaccard et al. (2013, Science) is a description of the surface-to-bottom exchange of waters (i.e. ventilation). Ventilation is however not what is measured by Nd isotopic compositions in exported AABW. Maybe it is in some indirect way, but it would be great to hear how the authors make the link between the two. Another question I have is how we know about the export of AABW (i.e. flux). Do the authors infer an intrinsic coupling of export to the ventilation observed within the Southern Ocean? This point is touched on in the second last paragraph of the paper, but needs to be better developed.

(3) Tipping points

Using the terminology of tipping points implies abrupt changes from one state to the other. Is this justified for the system? Can this potentially be visualised by some cartoons that summarise the vision of the authors for these tipping points? (i.e. resumption of NADW in the north at each of the interglacials – driven by what process?; surface to bottom ventilation in the Southern Ocean during recent interglacials, but not during lukewarm interglacial – driven by what?).

(4) AAIW and AABW

Line 129 brings up strong AAIW formation (along with a sustained Agulhas leakage) as a mechanism for sustained strong NADW formation. This is not really developed any further. A few more sentences and a discussion in the light of existing AAIW records would be warranted. In the same paragraph the point is made that AABW flow and ventilation do not need to be coupled (see my point above). It is therefore concluded that AABW flow was strong during all of the interglacials of the past 800,000 years, including the lukewarm interglacials. What is this based on? The fact that absolute Nd isotope numbers do not change? How does this go together with changed G-IG Southern Ocean endmember compositions for AABW and the mechanisms how they come about?

Detailed points:

(i) There is a problem with the referencing in lines 66 to 70, as 11 and 17 refer to the same paper. Maybe number 17 is meant to refer to the paper by Bohm et al (2015).

(ii) Methodology: I am not fully following the treatment of uncertainty for the Nd isotope

measurements. Is the reported two sigma uncertainty a standard deviation? From two measurements of the bracketing standards? How were internal and external errors combined?

Reviewer #3 (Remarks to the Author):

I have reviewed the paper "Independent tipping point for North Atlantic and Antarctic overturning circulation" by Drs. Howe and Piotrowski. The paper presents new Nd isotope data from the deep equatorial western Atlantic over the past 800,000 years, and the authors draw the conclusion that changes in the rate or volume of production of North Atlantic Deep Water were not responsible for the difference between earlier, "lukewarm" interglacials, and later, warmer interglacials. The data are of high quality, and the error bars are clearly shown in the figures. The authors make a strong case for their preferred interpretation of the data, but the writing needs extensive revision for clarity at the sentence level. I outline several examples below, but this is not an exhaustive list, and the whole paper needs to undergo editing for clarity. This isn't trivial, and indeed some of the errors and/or confusing sentences have the potential to confuse the authors' intended meaning about important concepts. If the writing can be sufficiently improved, I think the paper will be a very strong submission. The data and ideas are good, and the paper provides important context for an interesting and unresolved question about the different major overturning cells and their relationship to global climate.

Specific comments:

Line 11-12: Is this meant to be "lower during... than the subsequent five interglacials", or "as low during... as the subsequent five interglacials"? The difference matters for the interpretation of this sentence (the first sentence of the abstract). This is the first of many examples where two clauses of a sentence are not aligned with respect to the tense, singular vs. plural, or other aspects important to clarity and ease of reading.

Line 32-33: "more recent" feels unnecessary here- the interesting thing is the fact that they were warmer. Use a date range as in the previous sentence.

Line 51: Change "of" to "between".

Line 51: Define Antarctic Bottom Water as AABW if using abbreviation on next appearance.

Line 61: That the core top sample matches sea water values is good, but it simply shows that the method is recording (not "reconstructing") seawater Nd. We then assume that it is also reconstructing seawater Nd. The difference is important.

Line 61: What does "regular" refer to? The range of values? The timing of glacial cycles? Be specific.

Line 62: Change "in" to "during".

Line 65: That the range of values stays constant/similar doesn't say anything about the transitions between states. Saying that the data alone imply that the Atlantic underwent "similar transitions" between G/IG states could mean that the timing, mechanism, or other aspects of the transitions were the same, which is beyond the scope of these data. Again, be more precise with the wording.

Line 66: Do the authors mean G/IG states "typified by" the LGM and Holocene, rather than "typical of"? Or do they mean that there were many states typical of the LGM and Holocene?

Line 75: The sentence beginning "The disagreement is with..." is confusing and needs to be reworded.

Line 89: This is an inference/interpretation, not an "observation".

Line 91: "extant" should be "extent".

Line 96-100: This sentence is unclear/confusing.

Line 100-103: The lack of preservation peaks has been interpreted to be ****the result of**** poorer exchange...

Line 109: Insert "during" or similar between "different" and "the".

Line 118-126: The relationship of this paragraph to the rest of the paper needs to be made clear early on. This is the first mention of Agulhas leakage, and the reader needs a little context for why the authors switch to this topic. It doesn't become clear until the following paragraph.

Line 146: Change "has" to "have".

Figure 1: Is it possible to make the marker for site 929 black, at least on the right panel? Right now it looks like the site has a very different Nd value than surrounding seawater.

Line 183: Change "was" to "were".

Line 185: Change "was" to "were".

Reviewers' comments:

We are very grateful to all three reviewers for particularly helpful and constructive reviews. As such we have endeavoured to make all changes suggested by the reviewers as detailed below. We believe this has produced a paper with more robust conclusions and more thorough discussion of our results.

Reviewer #1 (Remarks to the Author):

Jake Howe and Alex Piotrowski present a beautifully resolved, very robust eNd record from the eastern tropical Atlantic (ODP site 929). The record is quite unique and certainly represents an important step forward in our understanding of changes in ocean circulation through time. The location of the core seems to robustly record temporal changes in the proportion of northern- vs. southern sourced waters and its core top value aligns well with water-column measurements.

The record reveals that the proportion of northern sourced water remained quite stable during the interglacials of the past 800 kyrs. This observation is used to infer that the deep ocean overturning circulation sourced in the North Atlantic was independent upon the state of Antarctic overturning in the Southern Hemisphere during the lukewarm interval (430-800 kyrs). Jaccard et al., 2013 had previously suggested that the relatively colder lukewarm interglacials were characterized by a reduced dynamic range of overturning, which according to the authors did not affect deep water production in the North Atlantic. As such, they propose that both deep convection end-members were characterized by independent tipping points. This is certainly an interesting hypothesis, but to my opinion the argument developed throughout the manuscript is somewhat overstated given the evidence at hand. At this stage the manuscript remain quite descriptive and would benefit from a much more inclusive discussion.

We thank Reviewer 1 for their thoughtful and helpful review. We have done our best to address all of the comments of Reviewer 1 as they were all well-reasoned points.

Major comments –

* tipping point – I find this formulation unfortunate and quite frankly confusing. Many studies (e.g. Ramstorf, 2002; Böhm et al., 2015) have shown that AMOC intensity was evolving along a continuum. The only notable change in AMOC regime was reported to be associated with Heinrich stadials (off mode). The core site was always bathed by both AABW and NADW in various proportions as beautifully illustrated by the downcore record. Moreover, the comparison between d18O and eNd (Fig. S2a) suggests a coherent correlation throughout the interval considered.

Jaccard et al., 2013 invoke a reduced dynamic range of Antarctic overturning during the lukewarm interglacials, which I interpret as representing a continuum as well. The much more subtle wording used in the abstract (l. 21-23) is much more convincing and appropriate.

In light of this comment and those of the other reviewers we have removed all reference to tipping points and now focus the paper on the decoupling between atmospheric CO₂ and NADW proportion. We have also changed the title of the paper to reflect the more subtle wording of the abstract as highlighted here.

Furthermore, Adkins, 2013 argues for a strong coupling between northern- and southern overturning circulation, in contradiction with the hypothesis presented here (e.g. l. 113-117). The argumentation would

generally benefit a lot from a more inclusive discussion on these two particular points.

We were very grateful to the reviewer for highlighting this discrepancy and have restructured our discussion in line with that proposed by Adkins 2013, who notes the coupling between northern- and southern overturning circulation as stated by the reviewers. Adkins states that the remaining discrepancy the northern- and southern hemisphere climate may then be due to the position of the fronts in the Southern Ocean as controlled by the westerly winds. This proposition is entirely consistent without datasets and the other datasets we compare it to form the literature, as such it is not our favoured interpretation of the decoupling between NADW proportion in the deep Atlantic and atmospheric CO₂ concentration during the lukewarm interglacials. Again we thank the reviewer for highlighting this point and this reference.

* I find it very surprising that the authors omit to compare their results with the Böhm et al., 2015 ODP1063 record covering the interval 0-150 kyrs. Both records share many similarities, but are also characterized by substantial differences, notably during MIS 5e and 6 (this also applies to the eNd results published recently by Deaney et al., 17). These discrepancies could be related to the (relatively) lower resolution of the record presented here, which could potentially undermine the conclusions presented in the manuscript.

We now compare the last 150 kyr of our record with the Bohm et al record as well as two records from the South Atlantic and the Indian Ocean. Furthermore we discuss at length the reasons for the similarities and differences when compared to the Bohm record (lines 61-69 and 82-93) we also explain why the discrepancies do not hamper the interpretation of our record, in particular the fact that the extreme unradiogenic values observed at site ODP 1063 are not indicative of the northern end-member composition for the rest of the Atlantic.

* Can the authors exclude a scenario for which NADW would be strongly reduced during the lukewarm interglacial and would be replaced by stronger Labrador Sea water (LSW)? One could imagine a switch in the way AMOC is being circulated during the lukewarm interval, during which LSW would provide a much larger contribution than today.

The more general point I want to make here is that, taken at face value, the data only robustly indicate that a variable proportion of LSW was present at the core site throughout the past 800 kyrs (given the quite unradiogenic values overall) without allowing making any inference about NADW (*sensu stricto*). I would thus recommend using the more generic term “northern-sourced waters” instead of NADW.

In order to acknowledge that a greater proportion of LSW in deep water in the Atlantic could indeed have altered the northern end-member composition resulting in equally unradiogenic values at site 929 during the lukewarm interglacials as more recent interglacials we have changed out terminology to northern-sourced waters where appropriate throughout. We also note, however, that although we cannot rule it out we consider this explanation as less likely to have occurred as it requires a significant change in water mass densities for which there is, to our knowledge, no reported paleoceanographic evidence of and as such we believe the simpler explanation of similar northern end-member is more likely.

Minor comments –

I. 29 – “... Pleistocene WERE not all...”

Corrected

I.82–85 & Fig. S2a – It would perhaps be a good exercise to distinguish the data from the lukewarm interval and see whether the eNd-d18O correlation is characterized by a different slope when compared with the last 430 kyrs.

Done

I.95-96 – “... reveals that atmospheric carbon dioxide concentrations variation DURING THIS PARTICULAR INTERVAL is unlikely to be related to Northern...”

Added this comment (lines 140-141) and also added a section towards the end of the paper to reiterate this point (lines 232-238)

I.109 – “... was different DURING the lukewarm...”

Added (line 156)

I.122 and throughout – “Agulhas” not “Agulhus”

Corrected all 4 occurrences

I.130 – AABW mostly forms through shelf processes and deep convection. The fraction of AABW flowing into the Atlantic Basin mostly forms in the Weddell Sea.

Sentence modified to include more detail, now mentions both shelf processes and deep convection explicitly (line 197)

I. 133-136. Very interesting argument!

Elaborated upon at request of other reviewers

I. 137 – eNd does not allow making any inference about flow speed/strength. Please rephrase.

Added explanation that evidence of flow speed comes from flow speed proxies, combined with our water mass provenance results suggest there must have been equivalent flow from the South Atlantic.

I. 146-152. The conclusion is overstated and largely speculative. I would recommend removing these few sentences.

Sentence toned down and final sentence removed (lines 239-244).

I. 179 - “are” not “ate”

Corrected

I. 279 - what type of relationship has been used to correct for the mass bias (0.7219) - exponential?

Yes, exponential mass correction, added to methods (line 423).

I. 282 – Are the authors referring to the standard error or the standard deviation?

Standard deviation – text edited to reflect this (line 426-430)

Reviewer #2 (Remarks to the Author):

In their paper 'Independent tipping points for North Atlantic and Antarctic overturning circulation' Howe and Piotrowski present a new high resolution record of authigenic neodymium (Nd) isotopes from the deep equatorial western Atlantic Ocean spanning the past 800,000 years. The authors interpret the reoccurrence of similar Nd isotopic compositions in glacial and interglacial periods throughout the record to indicate similar proportions of vigorous NADW flow at the site (i.e. full NADW resumption) at each of the interglacials. This observation is in turn taken to indicate decoupling of Northern Hemisphere processes from atmospheric carbon dioxide variations, which are 30-40 ppm lower during the lukewarm interglacials in the older parts of the record. In conjunction with a previously published productivity record from the Antarctic Zone in the Southern Ocean indicating reduced Antarctic overturning during lukewarm interglacials, the authors conclude that there are different tipping points for North Atlantic and Antarctic overturning circulations.

This new record is an exciting addition to develop our understanding on ocean circulation over the past 800,000 years and its role in setting atmospheric carbon dioxide concentrations. As such it should definitely be published. For my liking the paper however is too brief. As a consequence it leaves the reader a bit in the dark on the proposed mechanisms of 'independent tipping points for North Atlantic and Antarctic overturning circulation'. I will provide some examples below on points that could be further developed.

We thank Reviewer 2 for their constructive comments and have made our best effort to address all of them as detailed below.

(1) Quantitative statements on NADW flux

In lines 66 to 70 the authors use their own, previously published and carefully documented estimates for glacial endmember compositions of northern and southern-sourced waters to conclude that the new data from ODP Site 929 are consistent with lower NADW flux during glacial periods. This is in agreement with previous work (e.g. Bohm et al., 2015, Nature) but has not been demonstrated for the duration of the past 800,000 years and is hence an important finding. Suggested similar proportions of NADW during peak interglacials indirectly suggest that we are looking at two sets of endmember compositions for the Southern Ocean for modern and glacial conditions, which means that it is rather difficult to say something about the rigour or flux of water masses based on Nd isotopes alone. I am also wondering how the authors reconcile the apparent stability in northern endmember composition with their own findings of changed early Holocene NADW values in the abyssal ocean (Howe et al., 2016, Geology) in the light of such anomalous low values occurring at various points during the last glacial cycle (e.g. Boehm et al., 2015, Nature). I am not necessarily disagreeing on the point they make, but it would be great to see a more detailed discussion on how some of the interpretations can be reconciled in terms of modern and past Nd isotope data and endmembers.

In response to these concerns we have added a direct comparison between the ODP 1063 record (Boehm et al) in Figure 3 and an extensive discussion of the reason for the differences observed between the two records

(lines 61-69 and 82-93). This discussion details how the extreme unradiogenic values observed in the abyssal northwest Atlantic are not seen at mid depths (Howe et al., *Geology*, 2016) and as such are not likely to affect our site. Furthermore we expand upon the possibility of end-member change and the evidence against this proposition later in the discussion (lines 119-130) stating that high-resolution crust records from the mid-depth Atlantic show little evidence for end-member change in northern-sourced water across the past 500 kyr.

(2) North Atlantic and Southern Ocean overturning circulation

Along the same lines, it would be nice to get a bit more detail on what is actually meant by North Atlantic and Southern Ocean overturning circulations. Are we talking about the upper and lower limb of the overturning circulation in the way as defined by Ferrari et al. (2014, *PNAS*)? For the North Atlantic the answer is most certainly yes, but what about the Southern Ocean? The Antarctic overturning circulation in Jaccard et al. (2013, *Science*) is a description of the surface-to-bottom exchange of waters (i.e. ventilation). Ventilation is however not what is measured by Nd isotopic compositions in exported AABW. Maybe it is in some indirect way, but it would be great to hear how the authors make the link between the two. Another question I have is how we know about the export of AABW (i.e. flux). Do the authors infer an intrinsic coupling of export to the ventilation observed within the Southern Ocean? This point is touched on in the second last paragraph of the paper, but needs to be better developed.

We have considerably extended our discussion of the distinction between circulation strength/AABW flux and ventilation state of the Southern Ocean throughout the paper to make our assumptions both stronger and clearer. We make clear in lines 141 to 152 that the difference in the Southern Ocean during the lukewarm interglacials has been attributed solely to ventilation, not necessarily any change in strength of AABW formation. We elaborate on this point in lines 199-204 where we highlight that the strength of AABW production cannot be inferred from ventilation state, which is what the literature suggests was different in the Southern Ocean during the lukewarm interglacials compared to the more recent interglacials.

(3) Tipping points

Using the terminology of tipping points implies abrupt changes from one state to the other. Is this justified for the system? Can this potentially be visualised by some cartoons that summarise the vision of the authors for these tipping points? (i.e. resumption of NADW in the north at each of the interglacials – driven by what process?; surface to bottom ventilation in the Southern Ocean during recent interglacials, but not during lukewarm interglacial – driven by what?).

At the request of Reviewer 1 we have now changed the focus of the paper to not mention tipping points. Rather we now focus instead on the concrete observation that NADW proportion was decoupled from atmospheric CO₂ concentrations, and as such must have been decoupled from the ventilation state of the Southern Ocean, we also propose that the most viable physical mechanism to explain this decoupling is the position of the fronts in the Southern Ocean as controlled by the location of the westerly winds as suggested by others (Adkins 2013, Anderson, 2009).

(4) AAIW and AABW

Line 129 brings up strong AAIW formation (along with a sustained Agulhas leakage) as a mechanism for

sustained strong NADW formation. This is not really developed any further. A few more sentences and a discussion in the light of existing AAIW records would be warranted. In the same paragraph the point is made that AABW flow and ventilation do not need to be coupled (see my point above). It is therefore concluded that AABW flow was strong during all of the interglacials of the past 800,000 years, including the lukewarm interglacials. What is this based on? The fact that absolute Nd isotope numbers do not change? How does this go together with changed G-IG Southern Ocean endmember compositions for AABW and the mechanisms how they come about?

Added additional discussion with supporting published evidence to support the notion that AAIW production can be strong under cooler temperatures (lines 185-191) and thus there is no reason to conclude that AAIW production should have been weak during the lukewarm interglacials.

Yes, the reviewer's interpretation of our logic regarding AABW is the point we were trying to convey but we appreciate that it could have been more clearly explained so have expanded the discussion around that conclusion (lines 196-220). Although we think this is an interesting point it should also be noted we highlight that it is an indirect inference from a combination of paleo-proxy studies and the conclusion would be much stronger if derived directly from paleo-flow proxy studies. Furthermore this conclusion does not directly impact any of our main conclusions which are now focused instead upon the decoupling between Southern Ocean ventilation state and NADW production.

Detailed points:

(i) There is a problem with the referencing in lines 66 to 70, as 11 and 17 refer to the same paper. Maybe number 17 is meant to refer to the paper by Bohm et al (2015).

Citation to Bohm added.

(ii) Methodology: I am not fully following the treatment of uncertainty for the Nd isotope measurements. Is the reported two sigma uncertainty a standard deviation? From two measurements of the bracketing standards? How were internal and external errors combined?

Explanation has been expanded upon (lines 426-430 to make clear that the external error is the standard deviation of the bracketing standards from the relevant run session. The errors were combined as the square root of the sum of squares (line 429).

Reviewer #3 (Remarks to the Author):

I have reviewed the paper "Independent tipping point for North Atlantic and Antarctic overturning circulation" by Drs. Howe and Piotrowski. The paper presents new Nd isotope data from the deep equatorial western Atlantic over the past 800,000 years, and the authors draw the conclusion that changes in the rate or volume of production of North Atlantic Deep Water were not responsible for the difference between earlier, "lukewarm" interglacials, and later, warmer interglacials. The data are of high quality, and the error bars are clearly shown in the figures. The authors make a strong case for their preferred interpretation of the data, but the writing needs extensive revision for clarity at the sentence level. I outline several examples below, but this is not an exhaustive list, and the whole paper needs to undergo editing for clarity. This isn't trivial, and indeed some of the errors and/or confusing sentences have the potential to confuse the authors' intended meaning about important concepts. If the writing can be sufficiently improved, I think the paper will be a very strong

submission. The data and ideas are good, and the paper provides important context for an interesting and unresolved question about the different major overturning cells and their relationship to global climate.

We thanks Reviewer 3 for so effectively highlighting the shortcomings in our wording and have attempted to clarify not only all of the instances highlighted below but the use of such terms throughout the manuscript. We believe this has made for a much clearer manuscript that does not accidentally overstate the conclusions we draw from our data due to incorrect terminology.

Specific comments:

Line 11-12: Is this meant to be “lower during... than the subsequent five interglacials”, or “as low during... as the subsequent fiver interglacials”? The difference matters for the interpretation of this sentence (the first sentence of the abstract). This is the first of many examples where two clauses of a sentence are not aligned with respect to the tense, singular vs. plural, or other aspects important to clarity and ease of reading.

Should have been than instead of as. Corrected.

Line 32-33: “more recent” feels unnecessary here- the interesting thing is the fact that they were warmer. Use a date range as in the previous sentence.

Removed more recent

Line 51: Change “of” to “between”.

Changed

Line 51: Define Antarctic Bottom Water as AABW if using abbreviation on next appearance.

Defined as AABW

Line 61: That the core top sample matches sea water values is good, but it simply shows that the method is recording (not “reconstructing”) seawater Nd. We then assume that it is also reconstructing seawater Nd. The difference is important.

Changed reconstructing to recording.

Line 61: What does “regular” refer to? The range of values? The timing of glacial cycles? Be specific.

Changed to be specific “shows 100-kyr glacial-interglacial cyclicality”

Line 62: Change “in” to “during”.

Changed

Line 65: That the range of values stays constant/similar doesn't say anything about the transitions between states. Saying that the data alone imply that the Atlantic underwent "similar transitions" between G/IG states could mean that the timing, mechanism, or other aspects of the transitions were the same, which is beyond the scope of these data. Again, be more precise with the wording.

Changed phrasing to be as follows:

These glacial-interglacial ϵNd cyclicity at ODP site 929 over the last 800,000 years (Fig. 5a), suggest that overturning in the deep Atlantic underwent repeated transitions between similar glacial and interglacial states typified by the Last Glacial Maximum and the Holocene.

Line 66: Do the authors mean G/IG states "typified by" the LGM and Holocene, rather than "typical of"? Or do they mean that there were many states typical of the LGM and Holocene?

See above

Line 75: The sentence beginning "The disagreement is with..." is confusing and needs to be re-worded.

Change the sentence to start:

Our conclusions disagree with the findings of studies...

Line 89: This is an inference/interpretation, not an "observation".

Changed to inference

Line 91: "extant" should be "extent".

Changed to extent

Line 96-100: This sentence is unclear/confusing.

Broken in to two sentences for clarity (lines 143-149)

Line 100-103: The lack of preservation peaks has been interpreted to be **the result of** poorer exchange...

Added *the result of*

Line 109: Insert "during" or similar between "different" and "the".

Added

Line 118-126: The relationship of this paragraph to the rest of the paper needs to be made clear early on. This is the first mention of Agulhas leakage, and the reader needs a little context for why the authors switch to this topic. It doesn't become clear until the following paragraph.

Moved the definition of Agulhas leakage to the start of the paragraph so that it now links directly the end of the sentence before in that it discusses the control of NADW formation making the relevance to the rest of the paper immediately clear.

Line 146: Change “has” to “have”.

Changed

Figure 1: Is it possible to make the marker for site 929 black, at least on the right panel? Right now it looks like the site has a very different Nd value than surrounding seawater.

Changed

Line 183: Change “was” to “were”.

Changed

Line 185: Change “was” to “were”.

Changed

REVIEWERS' COMMENTS:

Reviewer #1 (Remarks to the Author):

I have now reviewed both the extensive response to the reviewers' comments as well as the thoroughly revised manuscript. The authors have done a great job in addressing the reviewers' comments in detail. The manuscript has improved quite substantially. I can only recommend this manuscript to be accepted for publication in NComms. Congratulations on this inspiring study!

I have one last very minor comment that I would like the authors to consider -

l. 161-162 – I don't necessarily agree with the second part of that sentence. If one considers the ODP1094 record at face value, the deglacial Ca/Fe peaks were subdued during the luke-warm interglacials, but not absent. In that sense, the resumption of NADW upon glacial terminations during this specific interval did result in somewhat better ventilation, although the Southern Ocean response was much smaller then.

Reviewer #2 (Remarks to the Author):

I enjoyed reading the revised manuscript by Howe and Piotrowski. Overall the authors did a very good job in considering the constructive criticism provided by three referees. As a result they produced a more rounded and better supported manuscript.

(1) The change in title is very appropriate and reflects that the discussion is now focussed on the decoupling of atmospheric CO₂ and NADW production during lukewarm interglacials, rather than controversial tipping points (i.e. addressing concerns from reviewers 1 and 2).

(2) It's also great to see that the authors now include a comparison with other published Nd isotope records from the deep North Atlantic, South Atlantic, and Indian Ocean (new Fig. 3). This really adds to the discussion and allows the reader to follow the arguments made more readily (i.e. addressing concerns from reviewers 1 and 2).

I have however one remaining point I would like to challenge the authors on. In response to reviewer 1, the authors argue that 'the extreme unradiogenic values observed at Site ODP 1063 are not indicative of the northern end-member for the rest of the Atlantic'. In my view it is important to consider that Site 929 lies at a similar water depth (4360m) and probably a similar neutral density level, but significantly further downstream along the flow path of the deep western boundary current from Site 1063 (4580m). Hence isopycnal (and diapycnal) mixing could readily explain the ~2 epsilon difference observed between the two sites, similarly to the additional ~2 epsilon units difference between Site 929 and the Cape Basin site. Following this argument of comparing sites along the same density contours in the modern ocean., one would think that the unradiogenic values at Site 1063, irrespective of their origin from deep Labrador seawater convection (Deaney et al., 2017) or boundary exchange in the Labrador Sea (Howe et al., 2016), should propagate down the western boundary current and get 'diluted' along its flow path. Isn't this exactly what is observed? Or is my assumption wrong that ODP Site 929 lies on the same density contour as ODP Site 1063 in the modern ocean?

(3) The authors did a good job in clarifying their view on the relationship between AAIW and AABW production and flux. I agree that more (and other) proxy evidence to constrain this point would be desirable, but does simply not exist right now.

Lines 39-42: At mid latitudes the atmosphere is the main player in transporting heat. The ocean can only really keep up in the tropics.

Reviewer #3 (Remarks to the Author):

The authors have done a nice job clarifying the manuscript, both with respect to editing the text, and adding points of discussion. I don't have any further concerns, and applaud the authors on a very interesting study.

We are grateful to all three reviewers for their extremely constructive reviews throughout the review process. We have addressed their remaining concerns as outlined below.

Reviewer #1 (Remarks to the Author):

I have now reviewed both the extensive response to the reviewers' comments as well as the thoroughly revised manuscript. The authors have done a great job in addressing the reviewers' comments in detail. The manuscript has improved quite substantially. I can only recommend this manuscript to be accepted for publication in NComms. Congratulations on this inspiring study!

I have one last very minor comment that I would like the authors to consider -

I. 161-162 – I don't necessarily agree with the second part of that sentence. If one considers the ODP1094 record at face value, the deglacial Ca/Fe peaks were subdued during the luke-warm interglacials, but not absent. In that sense, the resumption of NADW upon glacial terminations during this specific interval did result in somewhat better ventilation, although the Southern Ocean response was much smaller then.

We thank the reviewer for highlighting this point and have rephrased the conclusion to highlight that the lukewarm interglacials did not see ventilation comparable to the later interglacials rather than saying it did not result in better ventilation than the glacial state.

Reviewer #2 (Remarks to the Author):

I enjoyed reading the revised manuscript by Howe and Piotrowski. Overall the authors did a very good job in considering the constructive criticism provided by three referees. As a result they produced a more rounded and better supported manuscript.

(1) The change in title is very appropriate and reflects that the discussion is now focussed on the decoupling of atmospheric CO₂ and NADW production during lukewarm interglacials, rather than controversial tipping points (i.e. addressing concerns from reviewers 1 and 2).

(2) It's also great to see that the authors now include a comparison with other published Nd isotope records from the deep North Atlantic, South Atlantic, and Indian Ocean (new Fig. 3). This really adds to the discussion and allows the reader to follow the arguments made more readily (i.e. addressing concerns from reviewers 1 and 2).

I have however one remaining point I would like to challenge the authors on. In response to reviewer 1, the authors argue that 'the extreme unradiogenic values observed at Site ODP 1063 are not indicative of the northern end-member for the rest of the Atlantic'. In my view it is important to consider that Site 929 lies at a similar water depth (4360m) and probably a similar neutral density level, but significantly further downstream along the flow path of the deep western boundary current from Site 1063 (4580m). Hence isopycnal (and diapycnal) mixing could readily explain the ~2 epsilon difference observed between the two sites, similarly to the additional ~2 epsilon units difference between Site 929 and the Cape Basin site. Following this argument of comparing sites along the same density contours in the modern ocean., one would think that the unradiogenic values at Site 1063, irrespective of their origin from deep Labrador seawater convection (Deaney et al., 2017) or boundary exchange in the Labrador Sea (Howe et al., 2016), should propagate down the western boundary

current and get 'diluted' along its flow path. Isn't this exactly what is observed? Or is my assumption wrong that ODP Site 929 lies on the same density contour as ODP Site 1063 in the modern ocean?

We agree with the reviewer completely on this point and thank them for highlighting the lack of clarity in our revisions. We have now elaborated on our intended interpretation (exactly what the reviewer suggests) so that it should be clearer to the reader that the more radiogenic values at ODP site 929 than ODP 1063 are due to mixing of the unradiogenic values along the western boundary current with both southern-sourced water and more radiogenic northern-sourced water from the mid-depth Atlantic (lines 92-106).

(3) The authors did a good job in clarifying their view on the relationship between AAIW and AABW production and flux. I agree that more (and other) proxy evidence to constrain this point would be desirable, but does simply not exist right now.

Lines 39-42: At mid latitudes the atmosphere is the main player in transporting heat. The ocean can only really keep up in the tropics.

Rephrased to reflect the fact that ocean contributes to heat transport rather than controlling it entirely, thereby also acknowledging the importance of the atmosphere in this process.

Reviewer #3 (Remarks to the Author):

The authors have done a nice job clarifying the manuscript, both with respect to editing the text, and adding points of discussion. I don't have any further concerns, and applaud the authors on a very interesting study.

Thank you.